

# Evaluation of phosphate rock as the only source of phosphorus for the growth of tall and semi-dwarf durum wheat and rye plants using digital phenotyping

Mikhail Bazhenov[1], Dmitry Litvinov[2], Gennady Karlov[1] and
Mikhail Divashuk[1,2]

[1] Laboratory of Applied Genomics and Crop Breeding, All-Russia Research Institute of
Agricultural Biotechnology, Moscow, Russia
[2] Kurchatov Genomics Center-ARRIAB, All-Russia Research Institute of Agricultural
Biotechnology, Moscow, Russia

## ABSTRACT

**Background:** Phosphorus nutrition is important for obtaining high yields of crop plants. However, wheat plants are known to be almost incapable of taking up phosphorus from insoluble phosphate sources, and reduced height genes are supposed to decrease this ability further.

**Methods:** We performed a pot experiment using *Triticum durum* Desf. tall spring variety LD222, its near-isogenic semidwarf line carrying *Rht17* (*Reduced height 17*) gene, and winter rye (*Secale cereale* L.) variety Chulpan. The individual plants were grown in quartz sand. The phosphorus was provided either as phosphate rock powder mixed with sand, or as monopotassium phosphate solution (normal nutrition control) or was not supplemented at all (no-phosphorus control). Other nutrients were provided in soluble form. During experiment the plants were assessed using the TraitFinder (Phenospex Ltd., Heerlen, Netherlands) digital phenotyping system for a standard set of parameters. Double scan with 90 degrees turns of pots around vertical axis *vs.* single scan were compared for accuracy of phenotyping.

**Results:** The phenotyping showed that at least 20 days of growth after seedling emergence were necessary to get stable differences between genotypes. After this initial period, phenotyping confirmed poor ability of wheat to grow on substrate with phosphate rock as the only source of phosphorus compared to rye; however, *Rht17* did not cause an additional reduction in growth parameters other than plant height under this variant of substrate. The agreement between digital phenotyping and conventionally measured traits was at previously reported level for grasses ($R^2 = 0.85$ and 0.88 for digital biomass and 3D leaf area *vs.* conventionally measured biomass and leaf area, single scan). Among vegetation indices, only the normalized differential vegetation index (NDVI) and the green leaf index (GLI) showed significant correlations with manually measured traits, including the percentage of dead leaves area. The double scan improved phenotyping accuracy, but not substantially.

Corresponding author
Mikhail Bazhenov,
mikhabazhenov@gmail.com

## INTRODUCTION

Phosphorus is one of the main macronutrients needed for crop plants for achieving full yield potential. While concentration of inorganic phosphate (Pi) in plants is 5–20 mM, even the most fertile soils rarely provide more than 10 μM of free Pi, with typical concentration of Pi about 2 μM (*Bieleski, 1973*). For phosphorus supplementation of crops, soluble and insoluble (poorly soluble) mineral fertilizers are used. Soluble phosphates applied as fertilizers, being bound by calcium, iron or aluminum, rapidly turn into insoluble forms with limited availability for plants, while excess of soluble phosphates can get into ground waters and further cause eutrophication of water bodies (*Carpenter, 2008*). Insoluble phosphate fertilizers are a cheaper and more environmentally friendly source of phosphorus compared to soluble phosphate fertilizers. This is, for example, phosphate rock (phosphorite) powder obtained by grinding natural phosphorites—sedimentary rocks containing minerals of the apatite group ($Ca_{10}(PO_4)_6(F, OH)_2$) together with a wide range of non-phosphate minerals. Thus, the ability of crops to recover phosphates from insoluble forms is of great benefit to both agriculture and the environment (*Osborne & Rengel, 2002*; *Iqbal et al., 2020*).

There are several ways for plants to adapt to low availability of soluble phosphates in the soil through enhancing their phosphorus uptake efficiency. Among them are acidification of rhizosphere by release of hydrogen ions and organic acids (citrate, malate) to dissolve phosphates (*McGrail, Van Sanford & McNear, 2023*), releasing of phosphatases to get phosphorus from organic compounds in the soil (*George et al., 2008*), increasing root surface area by enhanced root growth (*Sun et al., 2002*) or root hairs growth (*Baon, Smith & Alston, 1994*), and increase in expression of phosphate ion transporters (*de Souza Campos et al., 2019*). Under field conditions, spatial distribution of roots and colonization with arbuscular mycorrhiza fungi also play significant role in phosphorus uptake efficiency (*Ramaekers et al., 2010*). Also, the reuse of phosphorus by redistributing it from old tissues and organs to young ones, reduces the overall need for this macronutrient (*Huang et al., 2011*).

The genotypic differences for phosphorus acquisition and use efficiency exist both between and within crop species (*Iqbal et al., 2020*). Wheat plants are considered to have poor ability to acquire phosphorus from its insoluble forms (*Ubugunov, Enkhtuyaa & Merkusheva, 2015*). At the same time, some studies showed that all genotypes of wheat, triticale and rye are able to some small extent to utilize phosphorus from iron phosphate ($FePO_4$), nearly insoluble form, and the ability of different genotypes to utilize it varies greatly (*Valizadeh, Rengel & Rate, 2002*; *Osborne & Rengel, 2002*). Rye in general seems to be more efficient in uptake of phosphorus from insoluble forms than wheat or triticale (*Osborne & Rengel, 2002*).

Utilization of reduced-height (*Rht*) genes in wheat breeding since 1960s allowed to increase grain yield almost twice. New short-straw varieties, being more resistant to lodging, allowed application higher doses of nitrogen fertilizers. Also, at the expense of stem biomass, semidwarf plants better supply assimilates for the developing spike (*Flintham et al., 1997*). Since that time, gibberellin-insensitive reduced-height genes (alleles

*Rht-B1b*, *Rht-D1b*) became widely used in wheat breeding programs worldwide. However, these genes have some adverse effects on important plant traits, like coleoptile length and seedling vigor (*Khadka et al., 2021*), 1,000 grain weight (*Hayat et al., 2019*), tolerance to drought (*Jatayev et al., 2020*), *Fusarium* head blight resistance (*Srinivasachary et al., 2008*), and nitrogen metabolism (*Li et al., 2018*). Recent studies showed that wheat cultivars carrying reduced height genes produce smaller amounts of organic acids exuded by roots. It may reduce the ability of wheat plants to uptake phosphorus from its insoluble mineral forms (*McGrail, Van Sanford & McNear, 2021*). In pot experiment using natural phosphorus deficient soil, semidwarf and wild-type cultivars were found to differ in phosphorus utilization efficiency—the old tall cultivars produced 1.8 times more of dry biomass per gram of absorbed phosphorus than modern semidwarf cultivars. The modern tall cultivars also exceeded the modern semidwarf cultivars by phosphorus utilization efficiency and phosphorus acquisition efficiency in both, phosphorus deficient and phosphorus supplemented soils, however the differences were not statistically significant (*McGrail, Van Sanford & McNear, 2023*). However, the direct studies on the effect of *Rht* gene on the growth of wheat on insoluble phosphate as the only source of phosphorus using fully controlled artificial substrate have not yet been conducted.

In this study we aimed to test the hypothesis of low ability of semidwarf wheat plants to grow on artificial substrate containing water-insoluble mineral (phosphate rock powder) as the only phosphorus source using two near-isogenic wheat lines that are different in *Rht-B1* gene (alleles *Rht-B1p* also known as *Rht17* and *Rht-B1a*) (*Bazhenov et al., 2015*). This study was performed using the destructive conventional methods to obtain end point data and also using non-destructive digital phenotyping of the aboveground plant parameters that were collected continuously. High-throughput phenotyping tools and platforms are considered to be promising for speeding-up the plant breeding process and able to provide new data for plant physiology. Using non-destructive sensor-based phenotyping, the daily changes in plant growth and the traits of early developmental stages could be captured, which may correlate with yield or stress tolerance. The 'TraitFinder' system, equipped with two 'PlantEye' high-resolution multispectral 3D laser scanners (Phenospex Ltd., Heerlen, Netherlands), allows obtaining time-series of both geometry traits, such as leaf area, and spectral vegetation indices of plants under controlled environment. Previous studies have tested the PlantEye scanners under both greenhouse and field conditions using different crop species, and produced reliable calculated parameters of plants, which correlated well with destructive measurements (*Vadez et al., 2015*; *Maphosa et al., 2017*).

In this study, beside the examination the effect of *Rht-B1* gene on the phosphorus acquisition in wheat, we compared the values of digital phenotyping measurements at the end of the study with the conventional phenotyping end point data to estimate the coherence between digital and conventional methods. In addition, we tested whether expanding the view of plants for 3D scanners by performing repeated scan with 90 degrees turn of plants can improve the accuracy of digital phenotyping.

## MATERIALS AND METHODS

### Plants and growth conditions in pot experiment

The study was conducted on the basis of the Kurchatov Genomics Center-ARRIAB, All-Russia Research Institute of Agricultural Biotechnology.

Three plant genotypes were used in the experiment: the tall spring durum wheat cultivar LD222 (*Triticum durum* Desf.), its semidwarf near-isogenic line (NIL) carrying gibberellin-insensitive *Rht17* (*Reduced height 17*, synonym of *Rht-B1p*) gene (*Bazhenov et al., 2015*), and semidwarf winter rye (*Secale cereale* L.) cultivar Chulpan, carrying a dominant height-reducing allele of *Ddw1* (*Dominant dwarf 1*, synonym of *Hl, Humilis*) gene (*Kobylyanskij, 1988*; *Goncharenko et al., 2019*). Winter rye (*Secale cereale* L.) was included in experiment as presumably more phosphorus-efficient control genotype (*Osborne & Rengel, 2002*).

The plants were grown in sand in 520 cm$^3$ plastic pots (maximum sizes at the top edge 9 × 9 × 9.5 cm) with closed bottom to prevent drainage, single plant per pot, in growth chamber at 23 °C day, 18 °C night, 16-h photoperiod, photon flux density 112 µmol/m$^2$s. The nutrients were provided in soluble form as Hoagland solution without phosphorus (*Hoagland & Arnon, 1950*), prepared from individual ingredients (Sigma-Aldrich, St. Louis, MO, USA; protocol is in Table S1). The phosphorus was provided either as phosphate rock powder mixed with sand in quantity of 2 g per pot, or in soluble form of monopotassium phosphate (a mixture of $KH_2PO_4$ + $K_2HPO_4$, designated as K-$PO_4$) at concentration 1 mM (normal nutrition control), or was not supplemented at all (no-phosphorus control). The experiment was performed in six biological replicates (3 genotypes × 3 treatments × 6 replicates = 54 plants) with a completely random design of placement of pots. The pots were filled with rounded quartz sand (0.5–1.0 mm fraction) or a mixture of sand and phosphate rock powder (23% $P_2O_5$, grade A, Verkhnekamskie Fertilizers LLC, Russia) and moisturized with corresponding nutrient-containing solution up to 75% of full water-holding capacity (652 g of dry sand + 120 ml of solution). The seeds were placed in wet sand at 2 cm depth, 2 grains per pot. The average grain weight was 0.040 g for rye Chulpan, 0.049 g for LD222 and 0.043 g for LD222 NIL with *Rht17* determined by weighing 36 seeds before sowing. After full emergence the excessive seedlings were removed, leaving a single normal seedling per pot. The pots were watered with equal volumes of distilled water each other day at the beginning, and each day at the end of the experiment, when the evaporation by the plants became greater. To compensate for the difference in water evaporation between pots, once a week, instead of a regular watering procedure, the pots were equilibrized by mass with distilled water on scales maintaining 75% of full water-holding capacity, which provided good conditions for growth. Although there were no water leaks from the pots, additional equal volumes of nutrient solutions were added every 2 weeks to compensate for nutrient losses due to assimilation.

### Digital phenotyping

During growth the plants were assessed using the TraitFinder (Phenospex, Heerlen, Netherlands) digital phenotyping system. TraitFinder represents a system of two

multispectral 3D laser scanners PlantEye F500 installed at an angle one to another on a moving platform above the table, on which the experiment plants are placed within definite area units. The scanners generate the 3D cloud of points and measure reflectance of the plant surfaces in red (624–634 nm), green (530–550 nm), blue (465–485 nm), and near-infrared (720–750 nm) light. Being used with manufacturers software (Phena 1.0, HortControl 3.5) it outputs a series of plant measurements including both morphological (geometry) parameters, as 3D leaf area ($mm^2$), digital biomass ($mm^3$), leaf inclination angle, and spectral indices, including normalized differential vegetation index (NDVI), green leaf index (GLI), normalized pigment chlorophyll ratio index (NPCI) and plant senescence reflectance index (PRSI). The detailed description of these parameters is provided at https://phenospex.helpdocs.com/plant-parameters and summarized in Table 1 (*Rouse, 1973*; *Peñuelas et al., 1994*; *Filella et al., 1995*; *Merzlyak et al., 1999*; *Louhaichi, Borman & Johnson, 2001*).

The digital phenotyping measurements began on the 2nd day after full emergence and repeated two or three times a week (scans every other day or two) approximately the same time of day until the end of the experiment (34 day after emergence). Each time two scans were made; the second scan was made immediately after the pots with plants were turned 90 degrees around vertical axis remaining on their places. To prevent lodging of plants during 3D scanning, which could cause them to fall out of scan area, a supporting wire of red color was installed in each pot. During data processing, the points of supporting wire were excluded from the 3D cloud of points using filtering by hue and NDVI parameters. The weight of the wire was accounted during watering the pots by mass.

## Conventional phenotyping

At the end of the experiment three of six plant replicates were randomly picked and phenotyped using more traditional methods. After the last measurement on TraitFinder, on the 34 days after seedling emergence, the above-ground portion of the plants was cut, the number of tillers was counted, the leaf area was measured using an office scanner (Canon MF734Cdw) and ImageJ 1.53k software. A ruler and pieces of color paper of known area were used for calibration. Both total leaf area, including dry brown and yellow leaves present on plants, and green leaf area were measured. The border between green leaf and senescent (yellow or brown) leaf areas was made subjectively. After scanning, the above-ground plant mass was put into open paper boxes, fixed at 105 °C for 10 min to prevent mass losses due to respiration and then dried at 60 °C until constant weight in drying cabinet. The dry above-ground mass was weighted on laboratory scales with milligram accuracy. To determine the below-ground biomass, the sand with roots was gently removed from the pots into a basin, the roots were washed from the sand in water, blotted with paper towel, dried in paper boxes to constant weight as described above and weighted.

## Measurement of pH of the solution in growth substrate

After roots removal, the pH of water extract of the remaining bulk soil in each pot was measured. The substrate was mixed with distilled water at 1:5 ratio, shaken for 30 min, left
**Table 1 Digital phenotyping (D. Ph.) parameters ('traits').**

| Phenotyping parameter | Description | Calculation and comments |
|---|---|---|
| *3D leaf area* | The D. Ph. analogue of leaf area. | A sum area of elementary triangles modeling the plant surfaces and derived from a cloud of points generated by 3D scanners |
| *Plant height* | The D. Ph. analogue of plant height that is unaffected or only slightly affected by wind and other factors during measurements. | The average height of the top 10% of the elementary triangles in a unit (single plant in a pot in our experiment). |
| *Digital biomass* | The D. Ph. analogue of biomass that correlate with plant biomass. | Calculated as a product of plant height and 3D leaf area. It is measured in $mm^3$, and assuming approximately the same tissue density throughout the plant, volume is a substitute for mass. |
| *Normalized differential vegetation index (NDVI)* | The spectral index correlates with amount of green vegetation, plant health and ability to intercept photosynthetically active radiation (*Rouse, 1973*). | NDVI is calculated using reflectance in near-infrared light ($R_{NIR}$) and red light ($R_{Red}$) by Formula (1) per each 3D point and then averaged over the whole unit. $$NDVI = (R_{NIR} - R_{Red})/(R_{NIR} + R_{Red}) \qquad (1)$$ |
| *Greenness, or Green Leaf Index (GLI)* | The higher the green reflection component to other channels, the higher the index (*Louhaichi, Borman & Johnson, 2001*). | Represents the reflectance in green ($R_{Green}$) channel, compared to other two visible channels—red ($R_{Red}$) and blue ($R_{Blue}$). $$GLI = (2 * R_{Green} - R_{Red} - R_{Blue})/(2 * R_{Red} + R_{Green} + R_{Blue}) \quad (2)$$ |
| *Normalized pigment chlorophyll ratio index (NPCI)* | Varies between −1 and +1 and correlates with carotenoid/chlorophyll ratio. | NPCI determined by TraitFinder is calculated by Formula (3) (*Peñuelas et al., 1994*), which may be negative to NPCI used in some other studies (*Filella et al., 1995*). $$NPCI = (R_{Red} - R_{Blue})/(R_{Red} + R_{Blue}) \qquad (3)$$ |
| *Plant senescence reflectance index (PSRI)* | Is used to evaluate leaf senescence and fruit ripening, and is sensitive to carotenoid/chlorophyll ratio (*Merzlyak et al., 1999*). | PSRI is calculated using reflectance in red ($R_{Red}$), green ($R_{Green}$) and near-infrared light ($R_{NIR}$) by Formula (4). $$PSRI = (R_{Red} - R_{Green})/(R_{NIR}) \qquad (4)$$ |
| *Hue* | Characterizes the color of the plant. | A component of the hue—saturation—value (HSV) color space calculated for each point of the cloud of points (3D model of plant obtained by 3D scanner) and averaged over the plant. |

to set for 10 min and measured using pH meter (Starter 3100 Bench pH-Meter; OHAUS, Parsippany, NJ, USA) at 0.01 pH accuracy.

## Statistical analysis

The data of conventionally (destructively) measured traits were subjected to factorial analysis (ANOVA) with genotype, nutrition variant and their interaction taken as factors using Statistica 6.0 software. For leaves/roots dry matter ratio and leaf senescence fraction the *ln(x)* transformation of data was applied for normalization of distribution before analysis. After analysis, the plots of marginal (least square) means with 95% confidence intervals were built. For transformed traits, the reverse transformation was done for each

point on the graph. The Tukey's honest significant difference (HSD) test with $\alpha = 0.05$ was used to group the means into homogenous groups.

The digital phenotyping traits were analyzed by plotting time series of average values along with standard errors using Python 3 libraries 'pandas' and 'plotnine'. A useful rule of thumb for normally distributed variables is that the observed sample mean will be within one standard error of the true mean 66% of the time, two standard errors 95% of the time and three standard errors 99% of the time (Hall, 1997). Thus, non-overlapping one standard error intervals may be interpreted as significant difference of means at $\alpha = 0.05$.

For making graphics, the averaged data of two scans with 90 degrees turn (double scan) were used. Before calculating means and standard errors, outlying data were removed using three median absolute deviation boundary following the algorithm: (1) for each sample of six plants for each trait, nutrition variant, genotype and time point, the median absolute deviation from sample median was calculated; (2) for each plant, the ratio of deviation of individual value from sample median to the median deviation was calculated, and if this ratio greater than 3, the individual value (plant) was discarded from sample. Calculations of means and standard errors were carried out taking the new sample sizes. The number of removed outlying values varied between 0 and 2 for any six-plant sample with averages between 0.5 and 0.7 for different traits and between 0 and 1.1 for different time points (Data S1).

Pearson's correlation coefficients of digital phenotyping traits *vs.* conventional traits were calculated in Statistica 6.0. For comparison of single *vs.* double scan with 90° turn, the last scan values (single scan) or an average of the two scans before and after the turn (double scan) values were used. Coefficients of variation for digital phenotyping traits were calculated for each genotype*treatment*day point and averaged around all experiment. Regression analysis was done in Microsoft Excel 2016.

The data obtained in the experiment are available as Supplemental File (Data S1).

## RESULTS

### Conventional phenotyping: root dry matter (endpoint data)

*Influence of Phosphorus Nutrition:* The dry root mass of plants grown without phosphorus supply (w/o P) was very small (0.1–0.3 g) and increased statistically significantly on soluble phosphate (K-PO$_4$) for all genotypes (to 1.7 g for rye and 1.2–1.3 g for wheat lines). The dry root mass under phosphate rock conditions was intermediate between the w/o P and K-PO$_4$ for the rye (1.1 g, statistically significant difference with two other conditions, here and further $\alpha = 0.05$ is applied) and closer to the w/o P conditions for both wheat genotypes (0.3–0.4 g, the difference is only significant compared to K-PO$_4$).

*Influence of Genotype:* There was no statistically significant differences between wheat lines under all three growth conditions. The dry mass of root of rye was significantly greater than of both wheat lines under phosphate rock growth conditions, and significantly greater than tall wheat line LD222-*rht* under K-PO$_4$ conditions (Fig. 1A).
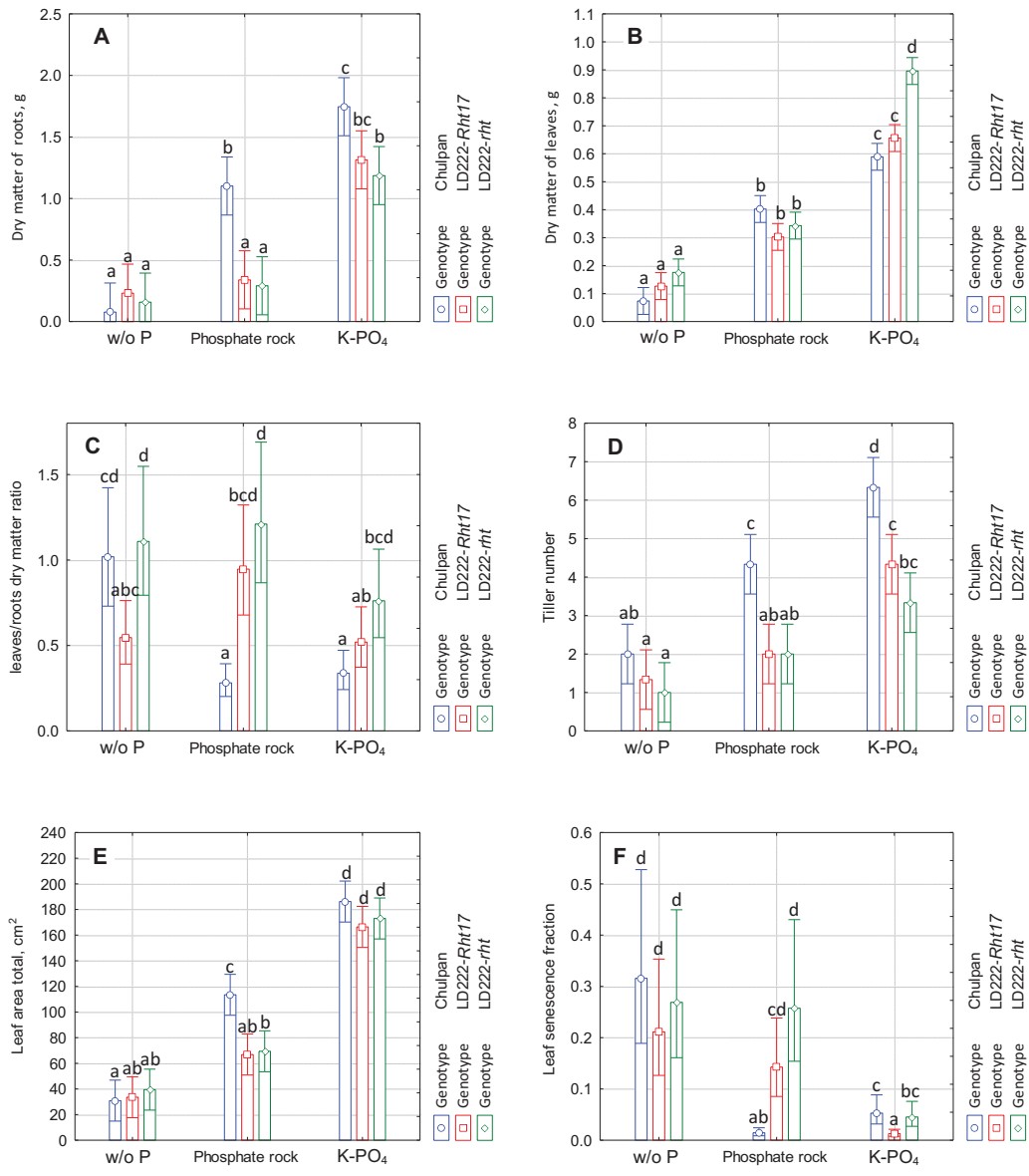

**Figure 1** **The diagram of means and 95% confidence intervals of the traits measured at the end of the experiment using traditional methods.** (A) Dry matter of roots. (B) Dry matter of leaves. (C) Leaves to roots dry matter ratio. (D) Tiller number. (E) Total leaf area. (F) Fraction of senescent leaf area. The letters above the bars designate the homogenous groups of means according to Tukey HSD (honestly significant difference) test at α = 0.05.

## Conventional phenotyping: leaves dry matter (endpoint data)

*Influence of Phosphorus Nutrition:* Similarly to dry matter of roots, the dry mass of leaves consistently increased in the series w/o P, phosphate rock and K-PO$_4$. The increase in leaves mass under phosphate rock was observed for all genotypes (from 0.08–0.18 g to 0.3–0.4 g, statistically significant for all genotypes). The leaves mass under K-PO$_4$ conditions was in the range 0.6–0.9 g and was statistically significantly greater than in other conditions for all genotypes.

*Influence of Genotype:* In contrast to the root mass, where the distinctive genotype was the rye, in case of leaves mass the distinctive genotype turned out to be the tall wheat line LD222-*rht*, which had statistically significantly different leaves mass only under K-PO$_4$ conditions (0.90 g *vs.* 0.60 and 0.65 g for rye and semidwarf wheat LD222-*Rht17* correspondingly). Under w/o P and phosphate rock growth conditions, there were no significant differences between genotypes (Fig. 1B).

## Conventional phenotyping: endpoint data on the ratio of dry matter of leaves to dry matter of roots (L/R)

The data reported above on the dry matter of leaves and the dry matter of roots on the last day of the experiment were used to calculate the ratio between them (Fig. 1C).

*Influence of Phosphorus Nutrition:* The only statistically significant effect was observed for rye that had the greater L/R ratio under w/o P conditions compared to both phosphate rock and K-PO$_4$ conditions (1.0, 0.3 and 0.4 correspondingly). There was no significant difference for any wheat line between any growth conditions.

*Influence of Genotype:* Under all growth conditions the L/R ratio was greatest for tall wheat line LD222-*rht* and the least for rye, with the only exception that under w/o P conditions the ratio for the rye was just slightly less than the ratio for LD222-*rht*. However, the variations in L/R ratio were relatively big, so the statistically significant differences between genotypes under the same growth conditions were observed only between LD222-*rht* and LD222-*Rht17* under w/o P conditions (1.1 and 0.5 correspondingly), between rye and tall wheat LD222-*rht* under K-PO$_4$ (0.4 and 0.8 correspondingly), and between rye and both wheat genotypes under phosphate rock growth conditions (0.3 and 0.9–1.2 correspondingly) (Fig. 1C).

## Conventional phenotyping: the number of tillers (endpoint data)

*Influence of Phosphorus Nutrition:* Among all genotypes, the number of tillers was always greatest for rye and the least for tall wheat LD222-*rht* except phosphate rock conditions when the number of tillers was the same for both wheat lines. The numbers of tillers per plant were statistically significantly different for rye under different nutrition conditions, and were 2.0, 4.3 and 6.3 for w/o P, phosphate rock and K-PO$_4$ correspondingly. Among wheat lines the only significant difference was in LD222-*Rht17* between phosphate rock and K-PO$_4$ (2.0 and 4.3 correspondingly).

*Influence of Genotype:* The number of tillers increased in the series w/o P, phosphate rock and K-PO$_4$ for all genotypes. The rye was the only genotype that had statistically significant differences compared to other genotypes. Under all growth conditions it had significantly greater number of tillers, both under phosphate rock conditions (4.3 *vs.* 2.0 in wheat lines) and under soluble phosphate conditions (6.3 *vs.* 3.3–4.3 in wheat lines) (Fig. 1D).

## Conventional phenotyping: total leaf area (endpoint data)

*Influence of Phosphorus Nutrition:* All genotypes increase leaf area in the series w/o P, phosphate rock and K-PO$_4$. For rye the difference between all growth conditions is

statistically significant (30, 115 and 165 cm$^2$ for w/o P, phosphate rock and K-PO$_4$ correspondingly). For both wheat lines the difference is significant only between K-PO$_4$ group and others (35–40, 65–70 and 165–175 cm$^2$ for w/o P, phosphate rock and K-PO$_4$ correspondingly).

*Influence of Genotype:* All genotypes show similar leaf area with a greater value for rye under phosphate rock conditions (115 cm$^2$ *vs.* 65–70 cm$^2$ for wheat lines; statistically significant) and, to some extent in soluble phosphate (185 cm$^2$ *vs.* 165–175 cm$^2$ for wheat lines; not significant) (Fig. 1E).

## Conventional phenotyping: leaf senescence fraction (LSF) (endpoint data)

On the last day of the experiment (34 days after seedling emergence), the area of senescence leaf was measured (Fig. S1), and the LSF was calculated (Fig. 1F) using total leaf area data presented above (Fig. 1E).

*Influence of Phosphorus Nutrition:* The greatest proportion of the senescent leaf area (yellow or necrotic leaves or their parts) was observed under w/o P conditions (LSF 20–30%). The least LSF was in K-PO$_4$ conditions (1–5%) for all genotypes with the only exception of rye on phosphate rock (1% *vs.* 5% and 21% on K-PO$_4$ and w/o P correspondingly). Statistically significant difference for LSF was observed between phosphate rock and K-PO$_4$ for all genotypes, and additionally for rye between phosphate rock and w/o P conditions.

*Influence of Genotype:* All genotypes were not significantly different by LSF under w/o P conditions. Under phosphate rock conditions the rye had significantly lesser LSF (1% *vs.* 15% and 26% for semidwarf wheat LD222-*Rht17* and tall wheat LD222-*rht* correspondingly). Under K-PO$_4$ growth conditions semidwarf wheat line LD222-*Rht17* was significantly healthier compared to rye and tall wheat line (LSF 1% *vs.* 4% and 5% correspondingly) (Fig. 1E).

## Dynamics of digital phenotyping parameters ('traits')

Digital phenotyping was carried out periodically from the second to the 34th day after seedling emergence. The dynamics of 3D leaf area is presented in Fig. 2 and of other digital phenotyping traits are presented in Figs. S1–S6.

*Influence of Phosphorus Nutrition:* 3D leaf area (Fig. 2) and Digital biomass (Fig. S2) changed in a similar way as the plants grew. Assessed by standard deviations, the statistical significance of differences in these parameters depending on the growth conditions appeared for rye after 7–10 days after germination, and for wheat lines after 10 days for K-PO4 compared with other conditions (after 17 days for Digital biomass for LD222-*rht*) and 17–20 days for the differences between phosphate rock and w/o P.

Under w/o P conditions the increase in Leaf area (Fig. 2) and Digital biomass (Fig. S2) occurred slowly and stopped in rye on days 17–20, and in both lines of wheat on day 27. Under K-PO$_4$ conditions, growth was most active and, unlike other conditions, was observed throughout the entire observation period, with the exception of the Leaf area parameter for LD222-*rht* that was stabilized at day 27. Phosphate rock provided a good
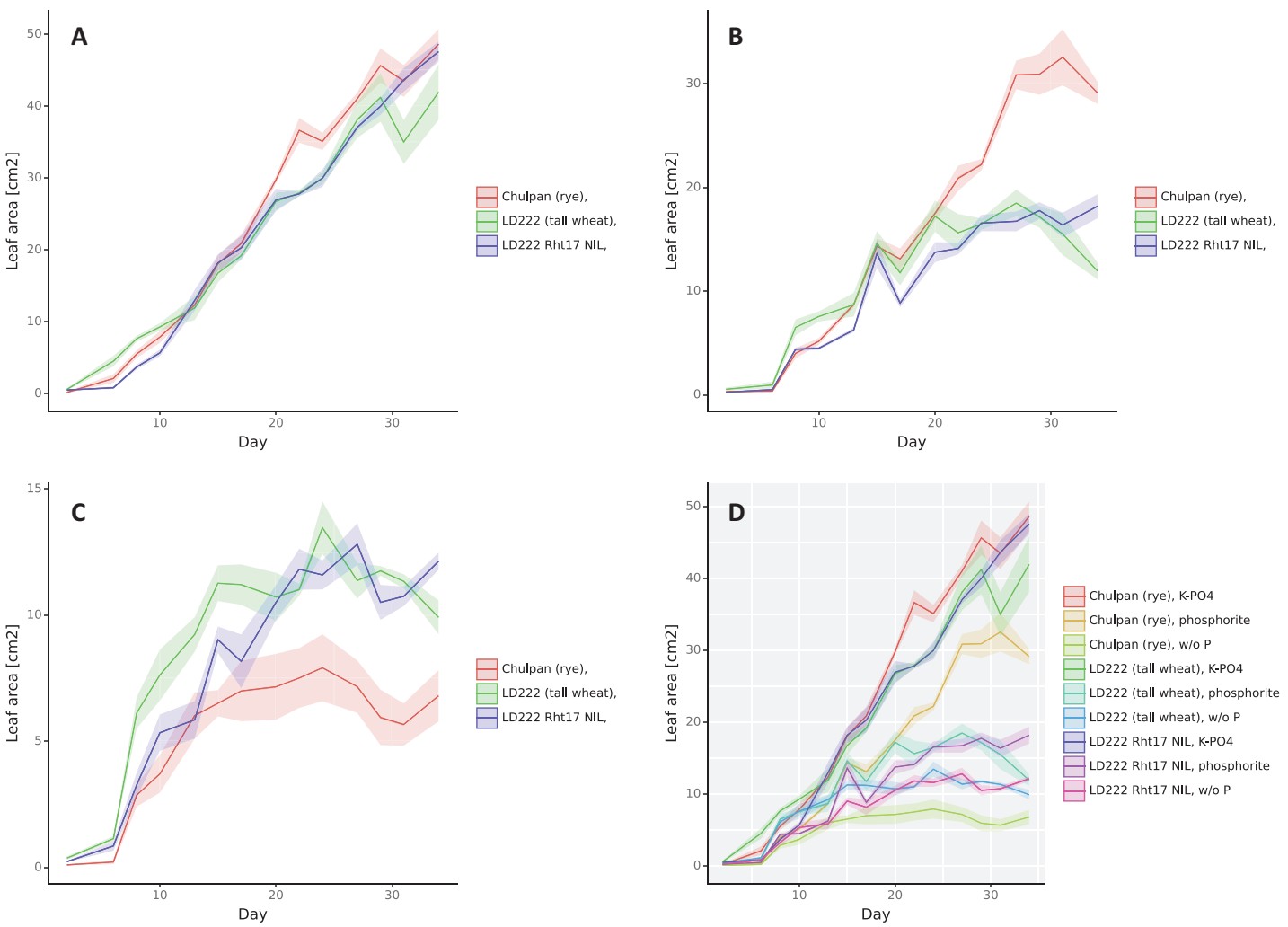

**Figure 2 The dynamics of leaf area (cm²), double scan with a 90° turn.** The means are represented by solid color lines. The standard errors are shown as faint-color ribbons above and below the lines. Non-overlapping one standard error intervals may be interpreted as significant difference of means at α = 0.05. The x axis indicates the day after emergence. Nutrition variants: (A) soluble phosphates, (B) phosphate rock powder, (C) without phosphates, (D) all variants together.

increase in Leaf area and Digital biomass for rye (more than 50% of the values compared to K-PO₄), to a lesser extent for LD222-*Rht17* (more than 50% of Digital biomass and less than 50% of Leaf area compared to K-PO₄) and even to a lesser extent for LD222-*rht* (less than 50% of Digital biomass and 50% of Leaf area compared to K-PO₄). The growth on Phosphate rock was observed until 27th day, then almost stopped for tall wheat line LD222-*rht*, stopped for the rye (Digital biomass stopped several days later—on the 30th day) and was reversed for semidwarf wheat line LD222-*Rht17*.

The Height (Fig. S3) was increasing the greatest in K-PO₄, in lesser extent in phosphate rock and the least for w/o P for rye. The difference became statistically significant from 20th day (by the estimation by standard deviations) when plants at w/o P conditions virtually stopped to grow. The phosphate rock rye stopped to grow on day 27. For tall wheat line LD222-rht the height became greater for K-PO₄ compared to other conditions from day 20

(estimated as statistically significant) while no consisted differences were observed between w/o P and phosphate rock plants. For semidwarf wheat line LD222-*Rht17* all three growth conditions resulted in approximately the same Height increase without consistent difference. For this line, the increase of the Height was gradually slowed and by the end of the observation period virtually stopped under all growth conditions.

All spectral parameters changed strongly during the first 10 days when the leaves were formed, so the data are considered after this period. The value of NDVI for healthy leaves with normal chlorophyll content is usually above 0.5, and the higher the better. The NDVI for rye was the greatest for K-PO$_4$ and the least for w/o P (all three conditions are accessed as statistically significantly different after 10–17 days) (Fig. S4). However, by the end of the experiment NDVI comes to 0.45 under all three growth conditions (no significant differences from the days 30–33). NDVI for both wheat lines behaves similarly but stays above 0.5 under K-PO$_4$ conditions until the end of the study (significantly different compared to w/o P and phosphate rock from days 13–15). NDVI of phosphate rock plants are slightly greater than w/o P plants in case of LD222-*Rht17* and in lesser extent for LD222-*rht* plants, however there is no significant difference on the last day of the study. The Greenness was always the greatest for K-PO$_4$ compared to other conditions (estimated as statistically significant), and there were no consistent differences between w/o P and phosphate rock conditions (Fig. S5). The normal range of NPCI and PSRI is −0.1 to 0.2 and −0.1 to 0 correspondingly. Under K-PO$_4$ conditions NPCI and PSRI values generally lies withing these ranges (Figs. S6 and S7). Under phosphate rock and w/o P the values are slightly lower with no clear difference between them.

*Influence of Genotype:* 3D leaf area under K-PO$_4$ growth conditions increased with small differences between genotypes (Fig. 2). Until the day 13 after emergence the greatest leaf area was in the tall wheat line followed by rye followed by the LD222-*Rht17* wheat line with the statistical significance of the differences within days 3–10 (as estimated by standard deviations). From about 13 days after emergence, the ranks of variants were rearranging. The greatest leaf area was observed in rye followed by semidwarf wheat line followed by LD222-*rht* line. The statistically significant differences were observed between rye and both wheat lines around day 24 and between LD222-*rht* and two other genotypes after day 30. Under w/o P growth conditions the rye grew the slowest (statistically significantly from the day 20). Tall wheat LD222-*rht* line grew the fastest until day 20 (statistically significantly different compared to LD222-*Rht17* line and rye within 7–15 days), then both wheat lines grew with similar dynamics, however on the last day leaf area of LD222-*Rht17* plants became statistically significantly greater than other genotypes. Under the phosphate rock growth conditions, the rye had the greatest leaf area (statistically significantly different from wheat lines from day 22). Among the wheat lines, the tall wheat line appeared to have slightly greater leaf area most of the time (statistically significantly greater than LD222-*Rht17* line on days 7, 18, 21), however by the end of the study, the semidwarf LD222-*Rht17* line showed statistically significant greater leaf area compared to the tall LD222-*rht* line (Fig. 2). Digital biomass dynamics under K-PO$_4$ conditions showed generally the greatest values for tall LD222-*rht* line followed by rye and semidwarf LD222-*Rht17* line (Fig. S2). After day 21 the difference between LD222-*rht* line and other

genotypes can be estimated as significant, except the day 32. Under conditions of w/o P variant, the digital biomass of rye was the lowest among all genotypes (statistically significantly from day 10). Digital biomass was generally the greatest for the tall wheat line, however in about half of all the time points the difference with semidwarf wheat line was not significant. Under phosphate rock growth conditions, tall wheat LD222-*rht* line had the greatest digital biomass until day 27 (statistically significantly different from other genotypes in most time points), then its biomass started to slowly decrease and the rye surpassed in digital biomass other genotypes. The semidwarf LD222-*Rht17* line had generally the least digital biomass, however unlike LD222-*rht* line, it continued to increase its biomass steadily and surpassed the tall wheat line on the last day of the study (Fig. S2). The height of the plants was the greatest for the tall wheat line LD222-*rht* and the least for the rye under all growth conditions (Fig. S3). The differences were assessed as statistically significant between all genotypes for the last 10 days of the study except the difference between rye and semidwarf wheat line LD222-*Rht17* under K-$PO_4$ conditions.

NDVI was the lowest for rye under all growth conditions (Fig. S4), although the difference between rye and wheat lines was statistically significant just for the second half of the observation period for K-$PO_4$ conditions, and around 15–20 days after the germination for other conditions. The differences between wheat genotypes were not consistent within the observation period. However, considering the last days of the study, the semidwarf line LD222-*Rht17* had statistically significantly greater NDVI than the tall wheat line in w/o P and K-$PO_4$ growth conditions, while no significant difference under phosphate rock conditions. The greenness was lower for rye for the most time points for all conditions compared to wheat lines (Fig. S5). Among wheat lines, tall LD222-*rht* generally had greater greenness values under all growth conditions, although there were some time points, when the opposite was observed. NPCI and PSRI indices were the greatest for tall wheat line and the least for semidwarf wheat line LD222-*Rht17* for all conditions (Figs. S6 and S7) with the greatest difference under w/o P conditions followed by phosphate rock (the differences are statistically significant from the days 10–12) and the least difference under K-$PO_4$ conditions (the differences are statistically significant from the day 22).

## The effect of plant rotation on the accuracy of digital phenotyping

Two variants of digital phenotyping of plants have been tested. The plants were scanned with TraitFinder phenotyping system and then scanned again after 90 degrees rotation. In the first variant of phenotyping ("single scan" without turns), digital phenotyping 'traits' from six replicates of plants with the same genotype and growth conditions were averaged (after removing outliers if they were determined) without taking in consideration the data obtained after 90° turn. In another variant of phenotyping ("double scan with 90° turn"), digital phenotyping 'traits' from each plant were average from two scans (before and after 90° turn) and these averages were used to average digital phenotyping 'traits' from six plant replicates. The coefficients of variations of the repeats turned out to be the same or only with slight improvement for double scan with 90° turn (Table 2).

**Table 2 Mean coefficients of variation (%) of digital phenotyping traits measured using single scan or averaging two scans with 90 degrees turn of pots around vertical axis.**

|  | Digital biomass | Plant height | 3D leaf area |
|---|---|---|---|
| Single scan | 29 | 10 | 27 |
| Double scan with 90° turn | 28 | 10 | 26 |

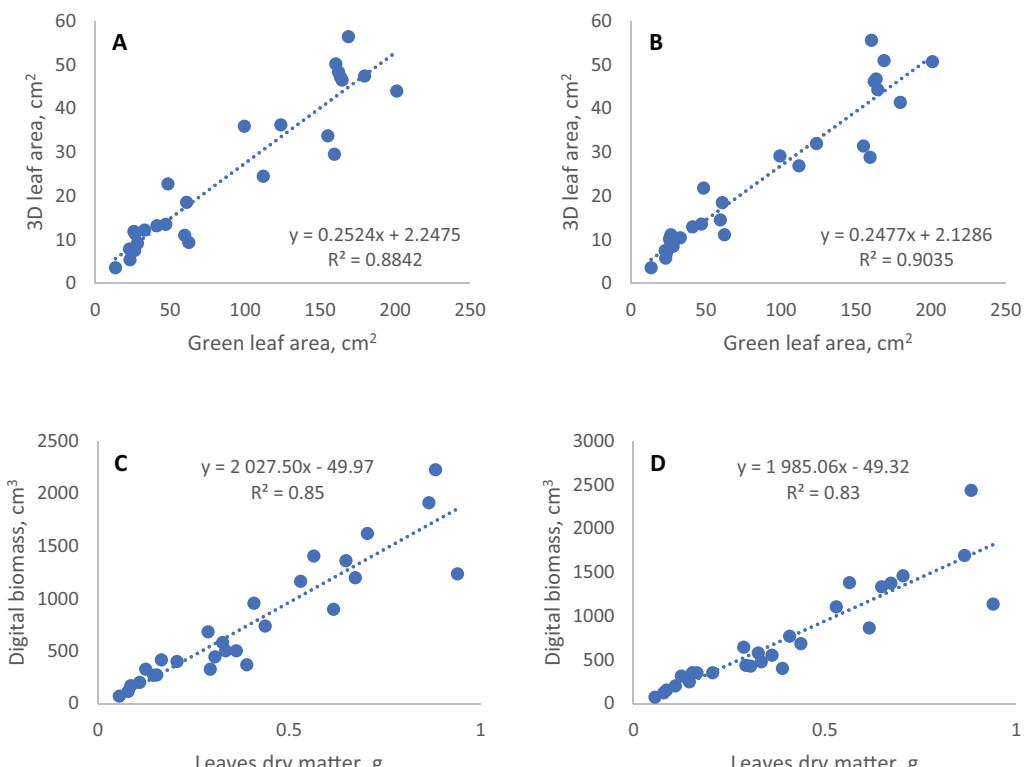

**Figure 3 Linear regression of conventional and digital phenotyping traits.** 3D leaf area measured by TraitFinder (A, single scan; B, double scan) *vs.* green leaf area measured by flatbed scanner; digital biomass measured by TraitFinder (C, single scan; D, double scan) *vs.* leaves dry matter measured on scales.

## Correlation of data obtained by conventional and digital phenotyping methods

To evaluate the agreement of digital phenotyping data and conventionally measured traits we performed correlation and regression analyses of the last time point in digital phenotyping time series and the data presented above (Fig. 3, Table 3). The most significant correlations of the digital phenotyping parameters ('traits') with the conventional traits were found between the 3D leaf area measured using TraitFinder and two traits measured using a conventional 2D scanner: (1) the green leaf area (total leaf area minus the area of senescence leaf), and (2) total leaf area, which includes yellowed and dead leaves. Pearson's correlation coefficients for these correlations were 0.94 and 0.93 correspondingly and the correlations were statistically significant. Dry aboveground mass

**Table 3 Pearson's correlation coefficients between traits measured destructively and digital phenotyping traits measured by means of 'TraitFinder' on the last day of the experiment.**

| | | Digital phenotyping traits | | | | | | |
|---|---|---|---|---|---|---|---|---|
| | | Digital biomass | 3D leaf area | Hue average | GLI average | NDVI average | NPCI average | PSRI average |
| Single scan | | | | | | | | |
| Conventional traits | Above-ground dry mass | 0.92* | 0.87* | −0.52* | 0.72* | 0.59* | 0.20 | 0.39* |
| | Tiller number | 0.63* | 0.81* | −0.09 | 0.32 | 0.43* | −0.06 | 0.08 |
| | Total leaf area | 0.88* | 0.93* | −0.45* | 0.65* | 0.59* | 0.14 | 0.36 |
| | Green leaf area | 0.88* | 0.94* | −0.41* | 0.64* | 0.62* | 0.08 | 0.32 |
| | Senescent leaf area % | −0.69* | −0.78* | 0.17 | −0.51* | −0.70* | 0.29 | −0.05 |
| Double scan with 90° turn | | | | | | | | |
| Conventional traits | Above-ground dry mass | 0.91* | 0.87* | −0.45* | 0.73* | 0.64* | 0.15 | 0.33 |
| | Tiller number | 0.63* | 0.82* | −0.09 | 0.32 | 0.47* | −0.01 | 0.11 |
| | Total leaf area | 0.87* | 0.94* | −0.37 | 0.65* | 0.64* | 0.10 | 0.29 |
| | Green leaf area | 0.87* | 0.95* | −0.33 | 0.63* | 0.66* | 0.06 | 0.25 |
| | Senescent leaf area % | −0.68* | −0.77* | 0.05 | −0.44* | −0.67* | 0.27 | 0.05 |

**Note:**
* Coefficients are significantly different from zero at $p < 0.05$.

of plants correlated strongly with digital biomass measured using 'TraitFinder'. Pearson's correlation coefficient was 0.92 and it was statistically significant. Double scanning of plants with 90° turn compared to single scanning without the turn did little to change the coefficients of correlation of manually measured traits with geometric digital phenotyping parameters (Digital biomass and 3D leaf area)—Pearson's correlation coefficients decreased or increased by no more than 0.01 (Table 3).

The coefficients of determination ($R^2$) of regression models of digital phenotyping *vs.* conventional traits for leaf area and biomass were 0.88 and 0.85 in single scan respectively, and 0.90 and 0.91 (with two outlier points removed) in double scan with a turn respectively (Fig. 3).

Among the spectral indices, only the NDVI showed statistically significant correlation with all the traits measured by traditional methods. Most significantly and negatively this index was associated with the percentage of yellowed or dead leaf area (senescent leaf area %). *Vice versa*, the senescent leaf area percentage was the most significantly associated with NDVI among other spectral indices. The green leaf index (GLI) showed the highest correlation with dry aboveground biomass, as well as with leaf area. The hue index was negatively associated with dry aboveground biomass (Table 3). Both PSRI and NPCI indices did not show a significant relationship with any of the traits determined by traditional methods, especially when double scan with a turn was applied. In general, the spectral parameters in our experiment were less related to the conventionally measured traits, including the proportion of the senescent leaf area.

## pH of the aqueous extracts of plant substrate

The diluted media washed out from the sand or sand with phosphate rock powder was slightly basic with pH in the range 7.0–7.8 (Fig. S1D).

## DISCUSSION

Releasing organic acids that could dissolve water-insoluble phosphates is one of the supposed mechanisms of enhancement of phosphorus acquisition by roots of agricultural plants. Previous study suggested, that wheat lines carrying reduced-height genes produce less acids in roots exudates (McGrail, Van Sanford & McNear, 2021). Thus, it can be assumed that semidwarf cultivars should be less able to acquire phosphorus from insoluble forms including phosphate rock powder.

In this study we performed a pot experiment to measure continuous and end point growth parameters of wheat near isogenic lines that differed by the gibberellin-insensitive reduced height gene alleles in the growth conditions of insoluble and soluble phosphorus supplementation. A rye cultivar Chulpan was used as a contrast genotype capable of utilizing insoluble phosphates. Beside morphological traits and parameters of physical growth of plants we also measured the spectral parameters as indicator of plant health. NDVI (Rouse, 1973) and GLI (Louhaichi, Borman & Johnson, 2001), both intended for differentiation of photosynthetically-active plant parts from non-living material, showed the most significant correlation with traditionally measured traits. Normalized pigment chlorophyll ratio index (NPCI) was proposed as highly correlated with the ratio between total carotenoids and chlorophyll (Peñuelas et al., 1994; Penuelas, Baret & Filella, 1995), reflects leaves yellowing as a result of nitrogen deficiency in wheat plants (Filella et al., 1995; Ranjan et al., 2012). The Plant Senescence Reflectance Index (PSRI) was found to be sensitive to the carotenoid/chlorophyll ratio and was intended to quantify leaf senescence and fruit ripening (Merzlyak et al., 1999; Anderegg et al., 2020). A feature of the spectral indices obtained using TraitFinder is that it takes into account only points belonging to plants, not soil. Thus, the index values reflect the condition of the green leaves and the proportion of dead or yellow leaves or similar areas in the green leaves.

For the plants grown on phosphate rock powder, the presence of Rht17, as expected, decreased the plant height with statistical significance in the last 10 days (Fig. S3), decreased digital biomass from day 10 to day 30 after emergence, and statistically significantly decreased spectral parameters: NPCI and PSRI since days 10–12 (Figs. S6 and S7), and greenness on most time points since day 18 (Fig. S5) determined by digital phenotyping. Also, under phosphate rock conditions there was a decrease in the leaf senescence fraction in semidwarf LD222-Rht17 wheat line compared to tall LD222-rht, although the difference was not statistically significant (Fig. 1F). At the same time, the presence of Rht17 did not change significantly any manually measured plant parameter at the end of experiment when plants were grown on phosphate rock (Figs. 1 and S1). On soluble phosphate, Rht17 decreased dry matter of leaves, leaf senescence fraction, plant height, digital biomass, greenness, NPCI and PSRI, and increased NDVI at the end point of measurements. The presence of this allele almost did not change the dry matter of roots, leaf area, and only slightly, statistically insignificantly, decreased leaves/roots dry matter

ratio. That is generally in agreement with previously studied effects of gibberellin-insensitive reduced-height genes (*McCaig & Morgan, 1993*).

The lower values of NPCI and PSRI spectral indices show either lower content of the pigments that absorb light in the blue (for NPCI) and green (for PSRI) range of spectra, such as carotenoids, or greater content of chlorophyll, or both, so the low NPCI and PSRI values are indicators, of healthier plants. NDVI is a proxy index for chlorophyll content, and it does not show any differences between wheat genotypes under phosphate rock conditions. Thus, under phosphate rock conditions semidwarf plants may produce less carotenoids than tall ones. For both wheat genotypes NPCI and PSRI values were not consistently significantly affected by the availability of phosphorus (Figs. S6 and S7). Also, lower values of these indices were observed for semidwarf wheat line under all three phosphorus supply variants, including growth on soluble phosphates. Thus, the differences in NPCI and PSRI between tall and *Rht17* wheat genotypes do not appear to be related to phosphorus availability. According to our results, retarded growth rather than changes in chlorophyll-carotenoids ratio or leaves yellowing, seems to be the main symptom of phosphorus deficiency.

Under soluble phosphate conditions, during the last week of the study semidwarf line LD222-*Rht17* had greater NDVI than tall line (Fig. S4). Since both wheat lines had lesser and equal NDVI values under phosphate rock conditions compared to soluble phosphate (Fig. S4), the decrease of NDVI can be considered as an indicator of phosphorus deficiency stress. Then the results of the growth under insoluble phosphate as compared to soluble phosphate is more negative for semidwarf wheat line than for tall line according to NDVI dynamics in the last week of the pot experiment. Also, the NDVI dynamics under insoluble phosphate was closer to the no phosphate conditions in case of semidwarf wheat, and closer to soluble phosphate in case of tall wheat during the last week of the experiment (Fig. S4).

For any genotypes, limited phosphorus supply retarded the plant growth (Figs. 1, 2 and S2) thus reducing consumption of other elements, that were supplemented equally in all variants. The decrease of NDVI during the last week of the study in case of rye and tall wheat under soluble phosphate conditions (Fig. S4) could indicate a limitation by other nutrients, such as nitrogen. Then the highest NDVI of semidwarf wheat line at the end of experiment in soluble phosphate conditions may indicate that it spends nutrients more modestly, keeping its tissues in good physiological state, while rye and tall wheat maintain vigorous growth and reutilizes lacking elements from old tissues. However, in the case of insoluble phosphate, the dry mass produced by plants is at least 50% less than in the case of soluble phosphate (Fig. 1), so there should be no nutrient limitation factor. The increase in biomass under phosphate rock powder conditions compared to the control without phosphorus was much greater in rye than in wheat. An especially significant increase was observed for root biomass. Thus, rye effectively acquires phosphorus from phosphate rock powder. This ability of rye plants to absorb phosphorus from insoluble forms could be due to presumably greater amount of acids in root exudates, other composition of acids (*Li, Ma & Matsumoto, 2000*), as well as higher root mass and visually observed higher root hair density (not measured in this study). Root-to-leaf dry mass ratio shows that rye allocates

more assimilates for the growth of roots, which makes it more adapted to acquire scarce nutrients from the soil. Better growth of rye compared to wheat on phosphate rock powder also may be explained by lower demand of rye tissues in phosphorus. Indeed, some studies show better phosphorus use efficiency of rye compared to wheat (*Pandey, Singh & Nair, 2005*).

Under no-phosphorus supply conditions the rye showed the least growth among all genotypes according to the plant height, 3D leaf area and the final biomass of leaves, while tall wheat line—the greatest growth and final biomass. In our experiment the grains of rye used for sowing were the lowest in 1,000 kernel weight, followed by semidwarf wheat line and tall wheat line in ascending order, thus positively correlating with final plant biomass. Previous experiment however, did not find significant correlation of seed phosphorus resources and growth of plants on limited P supply (*Osborne & Rengel, 2002*). The reserve of nutrients stored in the seeds should be kept in mind for element uptake experiments. In our case, the final differences between genotypes stabilized only after 20–25 days after seedling emergence.

So, by morphological traits, while the rye differed from wheat lines, we did not observe any negative effect of the *Rht17* gene on the growth of wheat plants on phosphate rock, the insoluble source of phosphorus. *Rht17* gene had the expected negative effect on the height of the durum wheat plants (Fig. S3) but did not significantly affect the root weight on any variant of phosphorus nutrition (Fig. 1A). This result is consistent with a recent study showing that the architecture of root system is not different between tall and semidwarf cultivars (*McGrail & McNear, 2021*). Also, phosphorus acquisition efficiency, that is, the amount of phosphorus absorbed per gram of root mass, was found to be equal for tall and semidwarf cultivars (*McGrail, Van Sanford & McNear, 2023*).

Earlier it was suggested that increased root surface area plays a more significant role than other traits in adaptation of plants to low phosphorus availability in the soil (*Bieleski, 1973*). Thus, it can be assumed that semidwarf wheat plants, despite less acid production by roots, can still be effective in phosphorus acquisition.

In our study, in addition to plant phenotyping, on the last day of the growing plants, the pH values of water extracts from the growth substrate (sand or sand with the addition of phosphate rock powder) were measured. The pH of medium in semidwarf and tall wheat lines shifted to higher values from 7.0 (neutral) for the control (no phosphorus) plants to 7.8 (slightly alkaline) for plants growing under insoluble or soluble phosphates supplementation (Fig. S1D). Under phosphate supplementation conditions the plant growth was promoted compared to the controls without phosphorus (Figs. 1B and 2). The greater rate of the growth requires greater rate of nitrogen assimilation, and the alkalization of the growth medium can probably occur due to the consumption of nitrate as the main source of nitrogen. It is known that nitrate assimilation cause alkalization of the soil (*Pierre, 1928*; *Weligama et al., 2008*, *2010*). Slightly alkaline pH is not favorable for dissolving of the phosphate rock powder (*Ellis, Quader & Truog, 1955*). Thus, although the difference in the pH was not statistically significant, we can assume that the absence of pH buffering or acidic components of natural soil in our growth substrate could obscure the effect of differences in the production of organic acids by wheat lines on their ability to

assimilate phosphorus from the insoluble source, the phosphate rock. Rye, however, maintained near-to-neutral pH of the substrate under all growth conditions (Fig. S1D). This may indicate that rye produces more acids in its root exudates.

Besides evaluation of the effect of semidwarf and tall alleles of the *rht* gene on wheat growth under conditions of insoluble phosphorus, two methodological questions of digital phenotyping were explored. The first question we explored was the consistency of digital and manually determined phenotyping data. The values of traits obtained by digital phenotyping, including leaf area, do not directly show the actual parameters of plants, but can be used to predict them through regression equations. However, the prediction of plant biomass and leaf area based on digital phenotyping data in our experiment with cereals has a rather large random error. The coefficient of determination ($R^2$) of linear regression models obtained using the data of single scan for digital biomass and 3D leaf area *vs.* observed traits are similar to ones obtained for grasses in earlier studies (about 0.86) (*Vadez et al., 2015*; *Maphosa et al., 2017*). Higher $R^2$ (about 0.95) were obtained in other studies for dicotyledonous plants having wider and more horizontally oriented leaf plates (*Kjaer & Ottosen, 2015*; *Vadez et al., 2015*). This large error of the geometric parameters of cereal plants could be due to predominantly vertical arrangement of the leaf blades in cereals, in the same plane with the scanning beam. This, together with leaf overlapping, can make the plants partially invisible to the 3D scanner, and the fluctuation of leaf position from one scan to another can produce this error.

The second question was whether the precision of digital phenotyping could be improved by repeated measurements of plants after rotation. Occlusion of some plant parts by others reduces the accuracy and precision of data in digital phenotyping. Second scan after 90 degrees turn theoretically may improve accuracy of phenotyping, making visible some parts of the plant, that were hidden for the scanners at the first scan. At the same time after the turn other plant areas become obstructed, and overall effect of the plant rotation was difficult to predict. Comparison of the coefficients of variation for six plants scanned at one position (no turn) and two positions (average of scans before and after 90 degree turn) showed that there was no reasonable improvement of precision on repeated scans (Table 2). To access the effect of the rotation on the accuracy of digital phenotyping, we assumed the manual data highly accurate and compared the correlation between digital and manually obtained data for single and double scan. With double scan the coefficient of determination $R^2$ increased from 0.88 to 0.90 for correlation of green leaf area and 3D leaf area, but decreased from 0.85 to 0.83 for correlation of leaves dry mass and digital biomass (Fig. 3). Thus, double scan did not provide consistent improvement in accuracy of digital phenotyping. This result is valid for plants whose appearance is similar to the plants we tested (1-month-old wheat and rye plants) and for a TraitFinder digital phenotyping system equipped with two PlantEye 3D scanners, such as the one we used in this study. The scanners are spaced about 50 cm apart and generate individual 3D point clouds, which are than merged into a single 3D plant model with lesser occluded areas than in the original scans. In case of systems equipped with a single 3D scanner, 90 degree turn can perhaps provide better effect on the precision of data.

## CONCLUSIONS

Gibberellin-insensitive dwarfing genes are widely used in wheat breeding. Previous studies concluded that these genes may compromise the ability of wheat plants to acquire and effectively use phosphorus from the soil. In our experiment using near-isogenic lines, we observed that gibberellin-insensitive reduced-height gene *Rht17* does not significantly reduce the growth of durum wheat plants on substrate containing phosphate rock (phosphorite) powder as the only source of phosphorus. Both tall and semidwarf durum wheat have shown a poor ability to acquire and use phosphorus from its insoluble forms. Drastically different from wheat, winter rye showed noticeable adaptation to the insoluble phosphorus source. Grown on sand supplemented with phosphate rock powder, it developed significantly greater root biomass and larger leaf area compared to the phosphorus-free substrate variant.

Introduction of digital phenotyping technologies is a modern trend in plant biology and agricultural research. The data obtained using the digital phenotyping systems are valuable for tracking plant parameters during their early growth without destruction of the experiment plants. Digital phenotyping may be relevant for experiments with a small amount of available material, for example, at early stages of the breeding process, in genetic engineering or plant physiology experiments. In our experiment, an application of repeated scan with a turn of plants for 90 degrees on TraitFinder (Phenospex) phenotyping system slightly improved accuracy of measurements. Therefore, in further experiments with cereal plants, it is sufficient to use the standard single-scan procedure without rotation.

### Funding

This research was funded by the Russian Science Foundation, grant number 21-16-00121. The funders had no role in study design, data collection and analysis, decision to publish, or preparation of the manuscript.

### Grant Disclosures

The following grant information was disclosed by the authors:
Russian Science Foundation: 21-16-00121.

### Competing Interests

The authors declare that they have no competing interests.

### Author Contributions

- Mikhail Bazhenov conceived and designed the experiments, performed the experiments, analyzed the data, prepared figures and/or tables, authored or reviewed drafts of the article, and approved the final draft.

- Dmitry Litvinov performed the experiments, authored or reviewed drafts of the article, and approved the final draft.
- Gennady Karlov conceived and designed the experiments, authored or reviewed drafts of the article, and approved the final draft.
- Mikhail Divashuk conceived and designed the experiments, authored or reviewed drafts of the article, and approved the final draft.

## Data Availability

The raw data are available in the Supplemental Files.

## Supplemental Information

Supplemental information for this article can be found online at http://dx.doi.org/10.7717/peerj.15972#supplemental-information.

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
