# Peer review of "Evaluation of phosphate rock as the only source of phosphorus for the growth of tall and semi-dwarf durum wheat and rye plants using digital phenotyping"

_PeerJ, doi:10.7717/peerj.15972_

## Round 0.1 · original submission · Major Revisions

Reviewers 1 and 2 raised several important issues. Please revise the whole manuscript as suggested by both reviewers. Some critical questions were raised by reviewer 2 which need a thorough explanation in order to better reveal the phenomenon that is captured by the manuscript.

Reviewer 1 ·

Basic reporting

The Estimating phosphate rock efficiency on the growth of tall and semi-dwarf durum wheat and rye plants by means of digital phenotyping is a well-written manuscript. The experimental design and methods which were used are appropriate and well-described. Results are clearly presented. The discussion econ needs to be improved. Namely, authors should give a better physiological meaning of the obtained results.

Experimental design

Please see the comments in the attached file.

Validity of the findings

Please see my comments in the attached file.

Annotated reviews are not available for download in order to protect the identity of reviewers who chose to remain anonymous.

Reviewer 2 ·

Basic reporting

Clarity of language could be improved in some cases. Professional English is used throughout with some minor typos. Literature incorporated into the introduction and, particularly, the discussion could be improved. Discussion is lacking in placing the results into context with other studies in many places. Some portions of discussion almost read as a reiteration of results. Study is missing a hypothesis.

Experimental design

Research question is defined but knowledge gap that is being filled could be more strongly identified. Methods for experimental portion of the study are mostly clear. Statistical methods need further description. I have a concern about the removal of data points based only on error. With the sample size and no other justification, removal of one let alone multiple data points is concerning.

Validity of the findings

Statistical interpretation of the data is lacking. P-values are not provided and statistical comparisons on graphs are not made. Without presentation of this evidence, readers are simply left with taking the word of the authors. Results and discussion will state there are differences without a comparison of values or indication of p-values. Conclusions could place study into a greater context.

Additional comments

Title should be revised. No measure of efficiency was calculated.

L42: High is used to refer to altitude and elevation. Suggest replacing high in “obtaining high yields” with improved or increased yields.

L43: Can you provide a range of general concentrations of plant available phosphorus? Batjes presents P limitations globally (https://doi.org/10.1111/j.1475-2743.1997.tb00550.x) and Bieleski’s review in Annual Review of Plant Physiology (1973, 24:225-252) provides common concentrations in different plant available pools.

L49: Cite the studies that support each of those mechanisms improving P uptake.

L63: The last name of the second author is Van Sanford not Sanford.

L67: McGrail, Van Sanford, and McNear published a recent paper that further explores the impact of dwarfism on P uptake efficiency that may be helpful for your literature review or discussion.

L64-66: Citation needed.

L68: This is an abrupt change that doesn’t flow with the rest of the paragraph. This topic should be explored in its own paragraph or moved elsewhere.

L86: Remove “tried.” You did estimate these traits.

L86: Place your study into a larger context. Why focus on the influence of semi-dwarfism in wheat and rock phosphate. I assume it is because rock phosphate is the primary source of phosphorus fertilizer. But why semi-dwarfism?

L93: What was the hypothesis for this experiment?

L106-107: Was the Hoagland solution made or purchased from a vendor? Provide information for the vendor or the chemical forms used to prepare the solution.

L113: Why was 75% water holding capacity chosen compared to 60%, which is generally regarded as optimal for microbial communities?

L116: Weighing rather than weighting.

L117-119: Can you add more information? I am confused by the meaning watered each day at the beginning and each day at the end of the experiment. Does this mean they were watered every day? I’m particularly confused because then in Lines 119-120 you write that water was added once a week.

L120-123: Revise for clarity. For example rather than writing “No leak of water..” At line 104 add that these were closed-bottom pots to prevent drainage. This will allow you to focus specifically on nutrient solution additions. For example, “To compensate for nutrient assimilation, additional equal volumes of nutrient solutions were added every 2 weeks.”

L123: Begin new paragraph.

L135-157: It may be beneficial to the reader to place these definitions into a table that you can reference during the discussion as needed.

L158-159: Repeated 2-3 times a week? Does this mean you spaced the measurements every three days which meant some weeks had two scans and others had three scans?

L161-164: This was a creative way to ensure the plants remained upright for imaging but didn’t influence the measurements!

L171: Plants not plats.

L171: “Green leaf area” or “area of green leaves” rather than “green leaves area.”

L170-171: Thoughtful detail to measure total leaf area which include brown/yellow leaves and then green leaf area.

L176-178: Were roots pulled as full systems and rinsed with water? Any sieving of the sand to collect additional root components?

L179-189: What statistical analyses were done to compare the plant indices collected — one-way or two-way ANOVA, T-test? How were differences separated — Tukey’s HSD? Was data normally distributed, or normalization was used? What was the level of significance — 0.05, 0.10?

L187-188: Was there another reason to remove these data points or than they were outside the three absolute deviations? How many data points were removed and what procedure was followed for their removal?

L192: Suggest retitling and dividing into two sections as “Influence of Genotype” and “Influence of Phosphorus Nutrition.” This is the results section that details differences in traits by genotype and phosphorus nutrition. Keep the first sentence in which you state that was done with conventional methods.

L197-234: Comparison of values needed and p-value of the analysis needed throughout. With phrases such as “slightly” or “differed little” I am assuming these are numerical differences rather than statistical. From Figure 1, there appears to be statistical differences but they are not indicated on the graphs with asterisks or letters.

L202: Excess is not the appropriate word. Increase in root mass or greater root mass is appropriate.

L205-217: Larger is used to refer to size. You want to use greater or increased to compare mass.

L212-213: This sentence belongs in the previous paragraph that discusses roots, not with a discussion of leaf mass.

L212-213: It is unclear what you mean by predominance of root mass. If you are ranking the root masses, use inequalities or direct comparisons. Because you have numerical measures of the root mass use these in your comparisons, not the phrasing weak or strong predominance.

L214-217: You are reporting positive values, but figure 1C shows negative ratios. The values reported do not match those on the figure. You are reporting them as root:shoot ratio, but the figure is number of leaves:root mass. Number of leaves:root mass is not the same as root mass:shoot mass.

L218: Higher is for a comparison of altitude or elevation. Greater is the appropriate word choice.

L193-239: I suggest reorganizing the results with all aboveground (leaf/tiller) traits and then root traits. It was a little confusing to read root mass, shoot mass, root:shoot ratio (which was really leaf number: root mass), and then a discussion of tiller number and leaf area.

L227-229: Revise for clarity. You are comparing two things in the same sentence which are two different observations: improvement of leaf area with a difference in phosphorus nutrition and leaf area between genotypes.

L242-265: One (L242-243) or two sentences (L262-265) is not a complete paragraph. Combine all lines into a single paragraph or expand on L262-265 so that it is a paragraph (at least 5 sentences) on its own.

L241: This is a comparison of the approach/method not traits. Retitle section as “Correlation of conventional and digital phenotyping methods.” Further you measured indicators of agronomic performance or plant characteristics not traits.

L253-254: The regression coefficient does not have to be 1, unless your hypothesis was that the two methods would approximate the same value.

L262-263: PSRI and NPCI didn’t correlate well with yellow/dead leaf area or green leaf area? No data present to help the reader evaluate this claim.

L268-281: In the discussion of these results data is missing to help the reader verify claims. What was the average and measure of error at the time points mentioned? Was there a statistical difference or a numerical difference? Reference Figure 3 immediately as you begin the discussion of these results. References Figure 3 at the end of the paragraph isn’t helpful for readers.

L268-333: There are many vague, broad generalizations throughout the results without data presented. Stating slightly smaller of almost equal areas of leaves, for example, requires data values to help readers interpret statements.The results, in general, are lacking in numerical support.

L336-337: You write previous studies but only cite one.

L337: The last name of the second author is Van Sanford not Sanford.

L341: What was the pH and might this have impacted the availability of P in the experiment?

L345: “Had” one more tiller not had got.

L345: Increased tillering was a breeding target with reduced height alleles (Lumpkin, 2015). If the stem length is decreased assimilated carbon can be used for other structures. Numerous physiological changes to wheat with Rht-B1b and Rht-D1b dwarfism included decreased leaf cell number (Keyes et al., 1989; doi:10.2135/cropsci1989.0011183X002900060023x), decreased leaf cell length (Keyes et al., 1989; doi:10.2135/cropsci1989.0011183X002900060023x), decreased leaf cell width (Jobson et al., 2019; doi:10.3389/fpls.2019.00051), decreased flag leaf length (Keyes et al., 1989 doi:10.2135/cropsci1989.0011183X002900060023), decreased flag leaf width (Jobson et al., 2019; doi:10.3389/fpls.2019.00051), increased leaf thickness (Nenova et al., 2014), decreased leaf blade area (McCaig and Morgan, 1993; doi:10.4141/cjps93-089), increased stomatal density (Nenova et al., 2014; doi:10.1111/jac.12090), increased stomatal frequencies of early leaves (McCaig and Morgan, 1993; doi: 10.4141/cjps93-089), decreased flag leaf photosynthetic rate, evapotranspiration, and stomatal conductance at anthesis (Jobson et al., 2019; doi:10.3389/fpls.2019.00051), greater radiation use efficiency from anthesis to maturity (Miralles and Slfaer, 1997; doi:10.1023/A:1003061706059), and increased yield through grain number with Rht-B1b and ear number with Rht-D1b (Borrell et al., 1991; www.jstor.org/stable/42758412 ) compared to tall, wild-type plants.

L345-347: Need a citation for the correlation to tillering and adventitious roots.

L354-356: Need a citation for these traits. Did you observe these traits or are you citing studies which support this possibility?

L362-363: How are you quantifying weak growth?

L365-370: You contradict yourself throughout these sentences. You state that rye and wheat are generally similar in terms of grain P but could differ up to 4-fold. You state differences in biomass could be from grain P which you did not measure. Then state there is no correlation of seed P and growth when P is limited.

L371-374: You did not measure P use efficiency. Shoots and/or roots were not analyzed for P content to calculate use efficiency, utilization efficiency, or uptake efficiency. While biomass production is dependent on adequate P, small differences in biomass do not mean there was a difference in P use efficiency.

L401: TraitFinder not TriarFinder.

L410-414: Leaf yellowing is symptomatic of P deficiency, particularly of older leaves first. The greater number of yellow leaves in optimum P nutrition could be nitrogen as you indicated or several other elements depending on the shape of the yellowing areas. It is also possible that the greater number of yellow leaves was because the lower leaves were no longer needed as the upper leaves became mature and were providing more assimilates through photosynthesis.

Figure 1: Line graphs are not appropriate here. Data is not connected over time in this figure. These are individual treatments means. Bar graphs of the means or box plots are appropriate.

Figure 2: Letters should be on the graph, not below it. Given large values in mm2, suggest converting to cm2. Use decimal point (period) instead of comma.

Figure 3: Recommend separating into a three panel figure with each of the P treatments as a panel. I’m not sure at what level the statistics were done and are important so this recommendation my not be the best separation of data. Separation of the figure into panels would also be helpful because some of the colors are similar. Because you are interested in the difference in leaf area over time and have several dats worth of data, I recommend a time-series analysis with an AR(1) structure so that data points one data collection apart (before and after) are compared for differences.

---

## Round 0.2 · accepted · Accept

Dear authors,
I am pleased to inform you that your manuscript has been accepted for publication. We appreciate and value your contribution to PeerJ.
Kind Regards
Naeem Khan

The Section Editor noted:

> Figure 1 would look better if the y-axis started at zero for all panels, instead of sometimes being below zero (since there is no negative data).

Reviewer 2 ·

Basic reporting

The revised draft meets all standards of basic reporting for PeerJ.

Experimental design

Author edits have addressed all questions about the methods and experimental design and have met the criteria for experimental design for PeerJ.

Validity of the findings

The authors have addressed all comments and concerns with respect to the validity of their findings.

Additional comments

The authors have made substantial revisions and addressed all comments and concerns. Their diligence has resulted in a much approved manuscript.